# Self-Reported Musculoskeletal Disorder Symptoms among Bus Drivers in the Taipei Metropolitan Area

**DOI:** 10.3390/ijerph191710596

**Published:** 2022-08-25

**Authors:** Yi-Lang Chen, Hans Alexander, Yi-Ming Hu

**Affiliations:** 1Department of Industrial Engineering and Management, Ming Chi University of Technology, New Taipei 24301, Taiwan; 2Department of Industrial Engineering, Krida Wacana Christian University, Jakarta Barat 11470, Indonesia

**Keywords:** logistic regression, musculoskeletal disorder symptoms, Nordic Musculoskeletal Questionnaire, risk factors

## Abstract

Bus driving is considered a highly stressful and unhealthy occupation, even among sedentary jobs, because of the particular task characteristics. This study used the Nordic Musculoskeletal Questionnaire (NMQ) to interview bus drivers and determine the risk factors for musculoskeletal discomfort. The NMQ was distributed to 152 bus drivers in the Taipei metropolitan area (Taiwan) and the valid data of 145 respondents were analyzed. The survey revealed that the overall prevalence of musculoskeletal disorder symptoms in any body part during the preceding year was 78.3%, and the body parts for which with the prevalence of discomfort was highest were the neck (46.9%), right shoulder (40.0%), lower back (37.2%), and left shoulder (33.8%). Stress and an uncomfortable seat may contribute to neck, shoulder, and lower back discomfort. Stretching between trips may help to reduce neck and shoulder discomfort. When comparing our results with those of similar studies, we discovered that the prevalence of symptoms and detailed risk factors vary by country and region. On this basis, we believe that local investigations emphasizing specific task arrangements and characteristics are needed to address the problem of musculoskeletal disorders in bus drivers.

## 1. Introduction

Musculoskeletal disorders (MSDs) are injuries or disorders of the muscles, nerves, tendons, joints, cartilage, or spinal discs that can cause sprains, pain, and inflammation [1]. Work-related musculoskeletal disorders (WMSDs) are MSDs caused or exacerbated by work or a work-related environment [2]. The UK Health and Safety Leader (HSE) revealed that in 2018 and 2019, 498,000 out of an aggregate of 1,354,000 business-related diseases were WRMSDs, a 37% predominance; furthermore, 29% of all lost working days were attributed to WMSDs [3]. Approximately 58% of the world’s population spends one-third of their lives working, and those who drive for a living are one of the most high-risk groups among all workers [4].

Bus driving is considered a highly stressful and unhealthy occupation because drivers must perform defensive driving tasks; take into account the safety of passengers on the bus, the safety of the vehicle, and the safety of other road users; and also comply with traffic rules and employer guidelines [5]. Moreover, professional driving requires extended periods of continuous driving [6]. Professional drivers must often deal with poor road conditions and have serious disadvantages, such as poor posture, time pressures, prolonged sitting, experiencing vibrations transferred from the vehicle–road interface, excessive physical exertion, and twisting motions during driving, which could challenge their health and cause a WMSD [7,8,9]. An extensive literature review reported evidence of the prevalence and severity of MSDs among professional drivers. The reported prevalence of MSDs affecting various body regions is high, with MSDs in the lower back being the most prevalent, followed by those in the neck, shoulders, and upper back [3].

Because of the substantial risks associated with bus driving, investigators have paid much attention to the prevalence of WMSDs in this population. Szeto and Lam [7] revealed that 481 bus drivers in Hong Kong experienced discomfort, primarily in the neck, lower back, shoulders, and knee/thigh with the 12-month prevalence ranging from 35% to 60%. A study conducted in Malaysia by Tamrin et al. [10] discovered a high prevalence of musculoskeletal pain among bus drivers; 60.4%, 51.6%, 35.4%, 40.7%, and 29.3% of drivers reported lower back, neck, shoulder, upper back, and knee pain, respectively. Moreover, a follow-up study [6] revealed almost no changes in the prevalence of WMSDs among bus drivers within seven years. A separate study, conducted in Malaysia at approximately the same time, reported an even higher prevalence of shoulder (79.4%) and neck (66.4%) discomfort [11]. In Thailand, Kasemsan et al. [12] recruited 83 bus drivers for a 12-month study and discovered an 81.9% and 80.7% incidence of neck and back pain, respectively, during the study period. In India, the corresponding prevalence of WMSDs was 73%, 70%, 55%, and 47.5% for back pain [13], lower back pain, neck pain, and shoulder pain, respectively [14]. However, in a separate study conducted in India by Yasobant et al. [15], only 26% of drivers (N = 280) had musculoskeletal problems in the neck, 24% in the back, and 20% in the upper extremities. Similarly, a low prevalence has since been reported in other papers [5]. For example, a study conducted in Iran reported that bus drivers complained most about their lower back (33.3%), upper back (18.3%), and knees (15.0%) [16].

Research examining the prevalence of WMSDs in bus drivers has been conducted not only in Asia but worldwide. A study of 200 Nigerian bus drivers revealed that the highest prevalence of pain was that in the lower back (73.5%) and that the driving performance of up to 74% of drivers was affected by pain [17]. Another study reported a more severe situation of an 85.0% prevalence among 200 minibus drivers in the same area [18]. In Ghana, 78.4% of bus drivers (116/148) reported having a WMSD during the preceding 12 months, with the prevalence being highest for back pain (58.8%) [19]. However, a lower WMSD risk was reported in Pretoria, where approximately 43% of bus drivers complained of back pain [20].

As these conflicting study results demonstrate, WMSDs are complicated by nature. An in-depth exploration of the causal relationships between WMSDs and risk factors is necessary so that appropriate healthcare programs can be initiated to facilitate effective prevention and treatment [3]. Moreover, it should be noted that although the disadvantageous conditions of bus drivers have attracted global attention, the severities of WMSDs vary geographically based on the characteristics of workers’ tasks and the working environment. Accordingly, localized studies are required to help solve the respective WMSD problems. In Taipei City, Taiwan, people use public transportation for their daily commute. Connections between mass rapid transit stations (MRTs) and services for areas without MRTs are usually supported by buses. Buses compensate for MRT shortages and offer several advantages such as flexible routes and the freedom to stop at any time. However, bus drivers can develop WMSDs in various parts of their bodies while performing this necessary and intensive work. Accordingly, this one-year cross-sectional study examined the prevalence of WMSD symptoms in various body parts using a revised Nordic Musculoskeletal Questionnaire (NMQ) distributed to 145 bus drivers in the Taipei metropolitan area. The responses to this questionnaire were used to clarify task characteristics and identify the risk factors for WMSD symptoms.

## 2. Methods

### 2.1. In-Depth Interviews and Field Observations

To develop a revised NMQ for bus drivers, we conducted in-depth interviews with 5 bus drivers who had job tenure ranging from 1.5 to 21.0 years. The purpose of these interviews was to understand their working details and task arrangements. We also observed and recorded the whole route for each bus driver while they carried out their daily work, as depicted in Figure 1. The NMQ was then developed based on these data.

### 2.2. NMQ

Figure 2 shows the flowchart of the NMQ developed for bus drivers in the study. As shown in the figure, the NMQ was based on in-depth interviews with bus drivers, field observations, and literature reviews. The questionnaire was divided into 3 main sections. The first section collected basic personal information including individual factors and previous injuries. The second section concerned job characteristics including task arrangements, driving-related factors, and details on seats and assistive devices. These task-related questionnaire items were generally adapted from the questionnaire developed by the Institute of Labor and Occupational Safety and Health (ILOSH), New Taipei City, Taiwan [21]. The variables measured in the first 2 sections of the questionnaire combined with the results are introduced in the Results section.

In the third section of the questionnaire, the NMQ was used to investigate symptoms of musculoskeletal discomfort among the bus drivers and included two parts: (1) Have you at any time during the last 12 months had trouble (such as aches, pain, and numbness) in the body part? and (2) What about the status, treatment, and reason for these symptoms, and their impact on daily activities? The NMQ is a general questionnaire that classifies musculoskeletal discomfort and symptoms of disorders in accordance with 9 major sites. This tool enables researchers to discriminate among sites of discomfort and injury and closely examine symptoms specific to certain sites. A special map facilitating the identification of the particular body parts was used [22]. The NMQ can be either general and assess the whole body or used specifically for each body part. Takekawa et al. [23] concluded that the NMQ is the most effective instrument for identifying respondents with chronic or recurring lower back pain. Deakin et al. [24] found that this self-reported questionnaire has reliability ranging from 77% to 100% and validity ranging from 80% to 100%. The questionnaire was also determined to be suitable for use in the Taiwanese population [25,26,27].

The content validity and relevance of the revised NMQ were confirmed by 3 experts, one of whom was the senior executive at a motor transport company and 2 of whom were occupational health professionals [28]. Reliability was examined using a test–retest method (N = 20) and by calculating the Pearson product–moment correlation coefficient. The interval between the first and second tests was 2 weeks. The correlation coefficients (*r*) ranged from 0.853 to 0.968 for all questionnaire items, which were designed to elicit various answers at specific frequencies.

### 2.3. Participants

We recruited 152 full-time bus drivers from 3 motor transport companies in the Taipei area using convenience sampling. All bus drivers had at least 1 year of work experience. The NMQ was administered on a one-on-one basis between 15 January 2021 and 30 March 2021. Written informed consent was obtained from all participants, and the study protocol received approval from the Ethics Committee of Chang Gung Memorial Hospital, Taiwan.

### 2.4. Statistical Analysis

Statistical analyses were conducted using SPSS software (version 22.0; IBM, Armonk, NY, USA) with *p* < 0.05 indicating statistical significance. Questionnaire data were examined through descriptive statistics and logistic regression. The odds ratio (OR) was used to compare the relative odds of occurrence of certain variables. In addition, Spearman’s rho correlation was used to identify the relationships between WMSD symptoms and various body sites.

## 3. Results

### 3.1. Demographics and Task Characteristics

Among the 152 respondents, 7 who did not meet the recruitment criteria were excluded and valid data from 145 bus drivers were finally analyzed. Table 1 presents the basic data for the 145 bus drivers. Their mean (standard deviation) age, height, and body mass index of the participants were 48.5 (9.3) years, 170.2 (6.0) cm, and 78.1 (14.9) kg, respectively. Table 2 details the demographics and task characteristics of the 145 bus drivers based on the questionnaire items. Among these respondents, the proportion with an abnormal BMI was 37.2%, and most of the respondents were overweight (mean BMI = 26.8 kg/m^2^). The proportions of drivers who smoked (45.5%) and drank alcohol (39.3%) were relatively high. Moreover, 30.3% of the respondents reported consumption of refresher drinks. Of our respondents, 61.4% slept ≤ 7 h a day and 27.6% suffered from a chronic disease including hypertension (N = 30), diabetes (N = 11), and heart disease (N = 10). A total of 92 respondents (63.8%) had an MSD, of whom only 40 reported a full recovery.

The drivers’ task characteristics are also detailed in Table 2. The majority of respondents (86.9%) had more than 3 years of driving tenure and 80.0% worked more than 5 days per week. Approximately one-third of bus drivers reported never stretching between trips (34.5%). The proportions of drivers reporting being low in spirits, having mental stress, and holding back urine while driving were 67.6%, 71.7%, and 77.9%, respectively. To improve comfort, several bus drivers frequently used assistive devices such as seat and back cushions (38.6% and 20.7%, respectively), neck support (4.1%), and sunglasses (67.6%). More than half of the respondents frequently adjusted the back support and horizontal position of their seat, and approximately one-quarter reported that the seat was uncomfortable.

### 3.2. WMSDs and Risk Factors

Table 3 depicts the NMQ results. The prevalence of musculoskeletal discomfort during the preceding year among the respondents was 78.3%. The respondents reported discomfort primarily in their neck (46.9%), right shoulder (40.0%), lower back (37.2%), and left shoulder (33.8%); the percentages for all other parts of the body were less than 23%. The cohort data were also compared with data obtained from other populations in Taiwan [21]. Further analysis showed that the main symptom of discomfort in these body sites was aching pain (60.2–88.5%) that slightly reduced their ability to work (approximately 50%) and a quarter to one-third of the respondents reported that it occurred almost daily and the symptoms had existed for more than three years (74.1%). However, most respondents neglected the discomfort or left it untreated. The primary treatment, if sought, was massage practices. A high proportion of the respondents considered the discomfort to have been caused completely or partially by work (e.g., neck, 76.2%; shoulders, 81.5–83.7%; lower back, 87.0%), indicating that most respondents deemed their discomfort to be the result of bus driving.

Table 4 reveals the results of a logistic regression analysis conducted to identify the risk factors for discomfort in various parts of the body, and the results are further summarized in Table 5. The results indicated that personal habits, historical injuries, psychological stress, and seat comfort rating were the main causes of WMSD symptoms.

### 3.3. Relationship of WMSD Symptoms with Main Body Sites

Table 6 presents the correlations of the discomfort symptoms among the different body parts. The 4 main sites of discomfort were significantly correlated with each other, implying that when discomfort is experienced in one body part, it may also be felt in other body parts.

## 4. Discussion

In our analyses, the prevalence of musculoskeletal discomfort in any body part during the preceding year was 78.3%, similar to that in a previous study [19] but higher [5,7,15] or lower than the findings of other researchers [11,12,18]. The differences in the symptom prevalence among the studies can be explained by the differing task contents and working environments considered in each study. In Table 2, we presented the demographics and work-related characteristics of the 145 bus drivers in our cohort. These data can be compared and contrasted with data in other studies in other regions. Some of our variables are similar to those of previous studies, whereas some were modified to reflect the local situation; thus, differences in the variables can account for the diversity of the results.

Although the prevalence of discomfort may have varied, most complaints of pain from bus drivers still concerned the neck, shoulders, and lower back in various studies [5,6,7,10,11,12,13,14,18,20,29]. In the present study, the respondents reported discomfort primarily in their neck (46.9%), right shoulder (40.0%), lower back (37.2%), and left shoulder (33.8%). These results are somewhat in contrast to most study findings reporting lower back pain as the predominant symptom [6,10,14,17,18]. Joseph et al. [3] conducted a review of 56 related studies and discovered that the lower back was the body region for which musculoskeletal pain was most frequently reported, with the meta-prevalence being 53%. However, several studies have revealed that discomfort in the neck and shoulders of bus drivers is more prevalent than that in their lower back [11,12,15]. This may be due to the different driving variables among the geographical regions of interest such as bus and road conditions. The incidence of lower back pain increases with seating duration [30]; therefore, because bus driving requires lengthy periods of sitting, it is considered a highly stressful and unhealthy occupation compared with other sedentary tasks [5]. Bus driving is also characterized by the simultaneous performance of numerous frequent tasks while being exposed to vibration, which has been identified as one of the major risk factors leading to lower back disorders for professional drivers [31]. The present study investigated drivers working in the Taipei metropolitan area where the road surfaces are relatively flat and smooth, which may reduce vibration [32]. A recent study conducted by Hanumegowda and Gnanasekaran [33] indicated that vibration and road types (i.e., asphalt pavement or rough road) were considered vital risk factors associated with WMSDs. They also found that for buses on asphalt pavement at >60 km/h, the vibration level was higher compared to a lower speed. In contrast, the vibration level exceeded the exposure action value on rough roads at all speeds (20–60 km/h). In practice, the allowed speed limit for buses in the Taipei metropolitan area is lower than 60 km/h. Additionally, starting in 2004, the Department of Transportation of Taipei [34] stipulated that the maximum service life of buses should not exceed 8 years; thus, high-quality vehicles may also reduce vibrations.

With similar driving tasks, the findings of this study differed from those of previous NMQ surveys of other professional drivers. Yitayal et al. [35] investigated 294 taxi drivers in Ethiopia and found that the prevalence of low back discomfort within one year was 27.9%, which was lower than our findings. However, this prevalence in 1242 taxi drivers in Taipei was 51% [36], which was higher than our study. Investigations also found that knee and lower back discomfort were associated with daily driving time [37] and mileage [38], respectively. These phenomena were not observed among the bus drivers in this study. A recent survey of 259 professional drivers in Nigeria also indicated that the overall prevalence of discomfort within one year in any body site was 21.2% [39]. The results showed that among the types of driving operations, WMSDs can vary significantly depending on the task characteristics and region.

A comparison of the questionnaire results with the newest findings of a government survey (conducted by the ILOSH, Taiwan; Table 3) regarding WMSDs in various work environments indicated that because of the nature of their work, bus drivers are markedly more likely to experience discomfort in the neck, lower back, buttocks, and knees than workers in other industries [21]. On the basis of the ILOSH’s classifications for industries, bus driving belongs to the warehousing and transportation industry. As shown in Table 3, the prevalence rates of discomfort in the neck, shoulders, and lower back in the warehousing and transportation industry were indeed higher than those of other industries. However, the prevalence of neck and lower back discomfort in the bus drivers in the present study was higher even though our sample size was relatively small. This may be partially attributed to the differences in the task characteristics. Driving as an occupation involves using repetitive muscle force to perform various tasks such as steering, shifting gears, and applying the brakes in continuous repetition [40]. Additionally, bus drivers must stare fixedly at the road for prolonged periods, leaving them unable to regularly extend and rotate their spine, and they are also subjected to the other disadvantages of continuous sedentary driving [5].

Logistic regression analysis revealed that personal habits (smoking, drinking, and no stretching between trips) were associated with neck and shoulder discomfort, whereas seat adjustment and assistive devices were associated with lower back discomfort. Table 5 presented the summary risk factors for the four primary body sites where discomfort is experienced. In the table, being in low spirits, having mental stress while driving, and having an uncomfortable seat were strongly associated with WMSDs for the four main body sites. The high prevalence of psychological stress and distraction may be related to the fact that nearly one-third of the respondents were accustomed to drinking refresher drinks. However, it is difficult to clarify the actual causes and effects of the risk factors related to personal habits. Even if personal habits such as smoking and drinking are inhibited, there is no guarantee that the related WMSDs will be improved. Therefore, instead of classifying these variables as risk factors, it is better to classify them as phenomena, which may have hidden effects similar to psychosocial factors [41]. Conversely, many other risk factors can be reasonably confirmed. Based on our findings, we believe that company executives should aim to reduce the stress on drivers and address their lack of energy while driving as well as improve the comfort of the driving seat. Stretching between trips could be an effective method for reducing neck and shoulder discomfort and should be encouraged. It should be noted that the majority of respondents indicated that the symptoms slightly influenced their work abilities and the consideration of these symptoms occurred almost daily, but these symptoms were mostly neglected or left untreated. This could be attributable to their socioeconomic status [42] because if they need to work to ensure financial support for their families, they neglect the WMSD symptoms and also the associated disorders. Thus, they stop working only when it is impossible to continue, and these symptoms were also considered the price to pay for working. This could underestimate the real prevalence of WMSDs and prevent drivers from fully recuperating and even worsen the symptoms [27].

In the analyses, psychological factors (that is, low spirits and stress while driving) significantly affected all body parts in which discomfort was experienced. Drivers must perform defensive driving tasks, taking into account the safety of passengers in the bus, the safety of the vehicle, the safety of other road users, and the need to comply with traffic rules and company with guidelines [5]. A stressful work environment increases muscle tension, leading to biomechanical stress, reduced blood flow, and an accumulation of metabolites [43] and may also decrease a driver’s pain threshold or change their perception and attribution of symptoms [44,45,46]. Furthermore, several studies have suggested that work-related psychosocial factors and stress may play a role in the development of neck and shoulder pain, particularly in driving occupations [7,47], although the etiopathogenetic mechanisms are poorly understood [48]. In the present study, psychological stress was not only a crucial factor affecting neck and shoulder discomfort but also caused lower back discomfort. Bus driving in city traffic can be considerably stressful for some individuals. Bergomi et al. [49] explored work-related stress in bus drivers and found the important role that personality traits play as they are associated with both activation of the neuro-endocrine response during driving and with the drivers’ perception of stress. In particular, the neurotic and impulsive traits of bus drivers were associated with higher stress perception. They affirmed that the adequate consideration of individual factors, such as personality traits, is useful in reducing stress in professional drivers.

Table 5 presented the strong correlations between an uncomfortable seat and discomfort in the neck, shoulders, and lower back. These results correspond with the findings of Tamrin et al. [6], who reported that uncomfortable seats cause WMSDs, with the OR being 3.40, whereas the OR values ranged from 2.52 for the right shoulder to 9.81 for the lower back in our study. Nazerian et al. [50] also indicated that 57% of truck drivers were suffering from discomfort in their lower back region, and seat comfort was found to be highly correlated with discomfort in the neck, shoulders, and back. Notably, Tamrin and colleagues found that a lack of seat adjustability was also a cause of discomfort [6], whereas the present study discovered the opposite. Lower back discomfort was more likely to occur in bus drivers who frequently adjusted their chairs or used seat cushions; however, this may infer that the respondents with lower back discomfort were more likely to adjust their seats and use seat cushions, obtaining limited benefits by doing so. This contradiction must be clarified in future studies. Furthermore, the correlation analysis results presented in Table 6 reveal that when a driver has a symptom of discomfort in any one body part, the likelihood of discomfort occurring in other body parts is higher. Hence, among the 145 bus drivers who participated, some employees may have had a predisposition to WMSDs and the healthcare department of their company should pay attention to these employees.

Several limitations of this study should be highlighted. The NMQ survey was limited in that only 145 bus drivers in Taipei, Taiwan, were recruited. The cohort was relatively small compared with the population of approximately 4500 bus drivers on the payroll in Taipei, and a larger cohort should be recruited and examined for further validation. Additionally, when generalizing the study results to other bus driver populations, attention should also be paid to the revised NMQ used in this study.

## 5. Conclusions

This study explored the symptoms of musculoskeletal disorders in Taiwanese bus drivers working in the Taipei metropolitan area. The NMQ and logistic regression were used to analyze the prevalence of symptoms and the risk factors for discomfort or injury to various body parts. The prevalence of overall discomfort among bus drivers within one year was 78.3%, with the prevalence being the highest for the neck, shoulders, and lower back. The diversity in the results regarding WMSDs in bus drivers among the different studies could be attributed to the different task designs and characteristics in the various countries and regions. Based on our findings, we believe that stress and uncomfortable seats could contribute to neck, shoulder, and lower back discomfort and we recommend that drivers stretch between trips to reduce neck and shoulder discomfort. The results of this cross-sectional analysis can serve as a reference and focus more attention on those who perform bus driving tasks every day.

## Figures and Tables

**Figure 1 ijerph-19-10596-f001:**
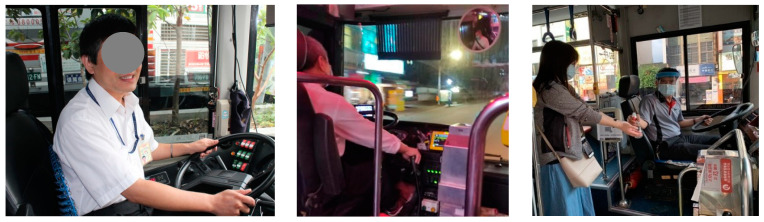
Bus drivers while working.

**Figure 2 ijerph-19-10596-f002:**
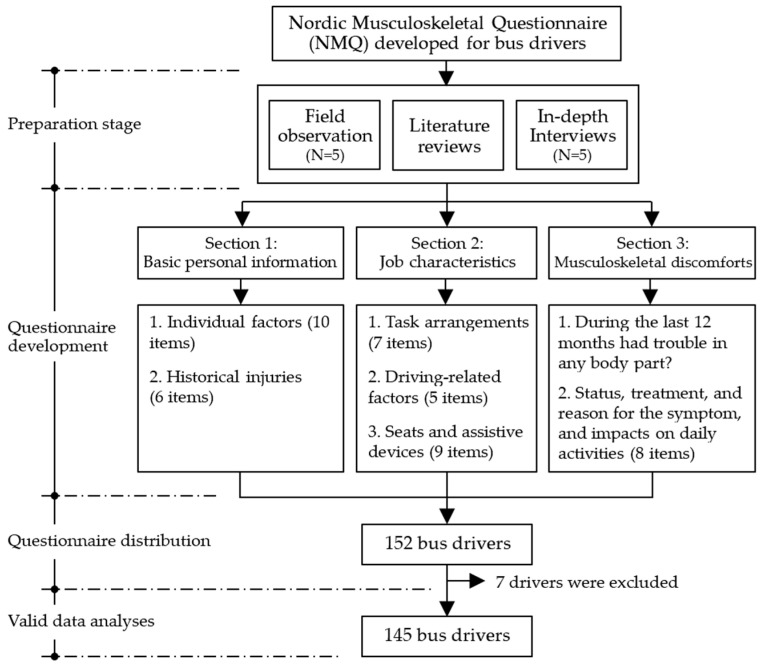
Flowchart of Nordic Musculoskeletal Questionnaire (NMQ) developed in this study.

**Table 1 ijerph-19-10596-t001:** Basic information on the 145 respondents.

Item	Mean	Standard Deviation	Median	Range
Age (years)	48.5	9.3	50.0	25.5–64.0
Stature (cm)	170.2	6.0	170.0	153.0–186.0
Weight (kg)	78.1	14.9	76.2	47.3–121.8
Body mass index (MBI)	26.8	4.6	26.5	17.7–39.4
Job tenure (years)	12.1	8.2	11.0	0.4–35.0
Workdays per week (days)	6.0	0.6	6.0	5.0–7.0
Trips per days	4.5	1.8	4.0	1.0–11.0

**Table 2 ijerph-19-10596-t002:** Demographics and task characteristics of the 145 respondents.

Variable	Category	N	Percentage (%)
**Individual factors**			
Gender	Male	130	89.7
	Female	15	10.3
Age	25–45 years	48	33.1
	>45–60 years	88	60.7
	>60	9	6.2
Body mass index (BMI)	Normal (18.5 ≤ BMI < 24)	91	62.8
	Abnormal	54	37.2
Dominant hands	Right	138	95.2
	Left	7	4.8
Exercise habits	Regular	50	34.5
	Sometimes or less	95	65.5
Tobacco smoking	Yes	66	45.5
	No	79	54.5
Alcohol drinking	Yes	57	39.3
	No	88	60.7
Freshener drinking	Yes	44	30.3
	No	101	69.7
Sleeping time	<7 h	89	61.4
	≥7 h	56	36.6
Chronic diseases	Yes	40	27.6
	No	105	72.4
**Historical injuries**			
Musculoskeletal disorders	Yes	92	63.8
	No	53	36.2
Neck and shoulders	Yes	54	37.2
	No	91	62.8
Lower back	Yes	39	26.9
	No	106	73.1
Wrist pain	Yes	16	11.0
	No	129	89.0
Knee pain	Yes	14	9.7
	No	131	92.3
Full recovery (from N = 92)	Yes	45	48.9
	No	47	51.1
**Task arrangements**			
Workdays per week	≤5 days	29	20.0
	>5 days	116	80.0
Driving route	Fixed	85	58.6
	Altered	60	41.4
Route area	Urban area	78	53.8
	Suburban	67	46.2
Shift	Need	96	66.2
	No need	49	33.8
Driving the same bus	Yes	125	86.2
	No	20	13.8
Trips assigned per day	≤4 trips	101	69.7
	>5 trips	44	30.3
Transmission type	Manual	62	42.8
	Automatic	83	57.2
**Driving-related factors**			
Job tenure	≤3 years	19	13.1
	>3 years	126	86.9
Stretching between trips	Never	50	34.5
	Occasionally or more	95	65.5
Low spirits when driving	Seldom	47	32.4
	Occasionally or more	98	67.6
Mental stress when driving	Seldom	41	28.3
	Occasionally or more	104	71.7
Holding back urine	Seldom	32	22.1
	Occasionally or more	113	77.9
**Seat and assistive devices**			
Seat cushion used	Seldom	89	61.4
	Frequently	56	38.6
Back cushion used	Seldom	115	79.3
	Frequently	30	20.7
Neck support used	Seldom	139	95.9
	Frequently	6	4.1
Wearing sunglasses	Yes	98	67.6
	No	47	32.4
Adjust seat-back support	Seldom	70	48.3
	Frequently	75	51.7
Adjust seat horizontal position	Seldom	36	24.8
	Frequently	109	75.2
Adjust seat height	Seldom	79	54.5
	Frequently	66	45.5
Adjust headrest	Seldom	120	82.8
	Frequently	25	17.2
Seat comfort rating	Comfortable	108	74.5
	Uncomfortable	37	25.5

**Table 3 ijerph-19-10596-t003:** Prevalence of musculoskeletal discomfort in various body parts for workers and workers in other industries (unit in percentage).

Body Parts	Entire Working Population (N = 17,757)	Warehousing and Transportation Industry (N = 698)	Recreation Industry (N = 179)	Other Service Industries (N = 656)	Bus Drivers in This Study (N = 145)
Neck	32.3	35.0	30.1	32.9	46.9
Shoulders	41.3	44.8	38.6	40.9	33.8/40.0
Upper back	22.3	27.3	20.8	23.8	20.7
Elbows	20.5	22.5	17.0	24.3	7.6/10.3
Lower back	31.0	34.4	29.2	30.7	37.2
Hands and wrists	26.5	27.3	28.1	30.5	11.7/14.5
Buttocks	11.8	14.4	14.4	12.7	22.1
Knees	16.9	19.0	15.2	20.4	21.4/22.1
Ankles	14.6	17.8	15.3	15.5	4.8/9.7

Notes: data obtained from ILOSH [21] and the present study; data with a slash mean left/right side.

**Table 4 ijerph-19-10596-t004:** Risk factors significantly associated with musculoskeletal disorders.

Body Parts (Prevalence %)	Risk Factors	Category	N	OR	95% CI
Neck (46.9%)	Stature	≤169 cm	81	1.00	—
		>169 cm	64	1.98 *	1.02–3.87
	Tobacco smoking	Never	79	1.00	—
		Occasionally or more	66	1.98 *	1.02–3.84
	Stretching between trips	Never	50	1.00	—
		Occasionally or more	95	0.09 ***	0.04–0.20
	Neck and shoulders injured	No	91	1.00	—
		Yes	54	3.69 ***	1.81–7.51
	Job tenure	≤3 years	19	1.00	—
		>3 years	126	3.87 *	1.22–12.31
	Low spirits when driving	Seldom	47	1.00	—
		Occasionally or more	98	3.89 **	1.80–8.38
	Mental stress when driving	Seldom	41	1.00	—
		Occasionally or more	104	14.80 ***	4.90–44.67
	Seat comfort rating	Comfortable	108	1.00	—
		Uncomfortable	37	3.71 **	1.66–8.30
Right shoulder (40.0%)	Tobacco smoking	Never	79	1.00	—
		Occasionally or more	66	2.75 *	1.39–5.45
	Alcohol drinking	Never	88	1.00	—
		Occasionally or more	57	2.38 *	1.20–4.73
	Stretching between trips	Never	50	1.00	—
		Occasionally or more	95	0.18 ***	0.09–0.39
	Neck and shoulders injured	No	91	1.00	—
		Yes	54	4.15 **	2.03–8.48
	Low spirits when driving	Seldom	47	1.00	—
		Occasionally or more	98	2.22 *	1.05–4.71
	Mental stress when driving	Seldom	41	1.00	—
		Occasionally or more	104	14.22 ***	4.13–48.98
	Seat comfort rating	Comfortable	108	1.00	—
		Uncomfortable	37	2.52 *	1.18–5.40
Lower back (37.2%)	Lower back injured	No	106	1.00	—
		Yes	39	5.57 ***	1.76–5.89
	Driving route	Fix	85	1.00	—
		Altered	60	2.54 **	3.19–9.83
	Job tenure	≤3 years	19	1.00	—
		>3 years	126	3.63 *	1.01–13.09
	Low spirits when driving	Seldom	47	1.00	—
		Occasionally or more	98	5.27 ***	2.15–12.89
	Mental stress when driving	Seldom	41	1.00	—
		Occasionally or more	104	2.71 *	1.18–6.25
	Holding back urine	Seldom	32	1.00	—
		Occasionally or more	113	2.54 *	1.02–6.37
	Seat cushion used	No	89	1.00	—
		Yes	56	3.13 **	1.55–6.32
	Adjust seat-back support	Seldom	70	1.00	—
		Frequently	75	3.47 **	1.70–7.11
	Adjust headrest	Seldom	120	1.00	—
		Frequently	25	3.12 *	1.28–7.56
	Seat comfort rating	Comfortable	108	1.00	—
		Uncomfortable	37	9.81 ***	4.11–23.44
Left shoulder (33.8%)	Tobacco smoking	Never	79	1.00	—
		Occasionally or more	66	2.54 *	0.98–6.54
	Alcohol drinking	Never	88	1.00	—
		Occasionally or more	57	4.69 ***	1.81–12.14
	Stretching between trips	Never	50	1.00	—
		Occasionally or more	95	0.14 ***	0.07–0.31
	Workdays per week	≤5 days	29	1.00	—
		>5 days	116	2.93 *	1.04–8.25
	Route area	Urban area	78	1.00	—
		Suburban	67	2.87 **	1.41–5.86
	Low spirits when driving	Seldom	47	1.00	—
		Occasionally or more	98	4.29 **	1.75–10.51
	Mental stress when driving	Seldom	41	1.00	—
		Occasionally or more	104	16.08 ***	3.69–70.11
	Seat comfort rating	Comfortable	108	1.00	—
		Uncomfortable	37	2.74 *	1.27–5.93

Notes: * *p* < 0.05, ** *p* < 0.01, *** *p* < 0.001; OR, odds ratio; CI, confidence interval.

**Table 5 ijerph-19-10596-t005:** Risk factors associated with WMSDs for 4 body parts.

Variable	Neck (46.9%)	Right Shoulder (40.0%)	Left Shoulder (33.8%)	Lower Back (37.2%)
**Individual factors**				
Stature	*p* < 0.05			
Tobacco smoking	*p* < 0.05	*p* < 0.05	*p* < 0.05	
Alcohol drinking		*p* < 0.05	*p* < 0.05	
Stretching between trips	*p* < 0.001	*p* < 0.001	*p* < 0.001	
**Historical injuries**				
Neck and shoulders	*p* < 0.001	*p* < 0.001		
Lower back				*p* < 0.001
**Task arrangements**				
Workdays per week			*p* < 0.05	
Uncertain route				*p* < 0.05
Route area			*p* < 0.01	
**Driving-related factors**				
Job tenure	*p* < 0.05			*p* < 0.01
Low spirits when driving	*p* < 0.01	*p* < 0.05	*p* < 0.01	*p* < 0.001
Mental stress when driving	*p* < 0.001	*p* < 0.001	*p* < 0.001	*p* < 0.05
Holding back urine				*p* < 0.05
**Seat and assistive devices**				
Seat cushion used				*p* < 0.01
Adjust seat-back support				*p* < 0.01
Adjust headrest				*p* < 0.05
Seat comfort rating	*p* < 0.01	*p* < 0.05	*p* < 0.05	*p* < 0.001

**Table 6 ijerph-19-10596-t006:** Correlations among body parts in which discomfort was experienced.

	Neck	Right Shoulder	Left Shoulder	Lower Back
Right shoulder	0.530	—		
	*p* < 0.001			
Left shoulder	0.614	0.548	—	
	*p* < 0.001	*p* < 0.001		
Lower back	0.248		0.204	—
	*p* < 0.01		*p* < 0.05	
Upper back	0.475	0.452	0.499	0.170
	*p* < 0.001	*p* < 0.001	*p* < 0.001	*p* < 0.05
Right knee	0.233	0.210	0.253	
	*p* < 0.01	*p* < 0.01	*p* < 0.01	
Left knee		0.261	0.232	
		*p* < 0.001	*p* < 0.01	

Notes: data are presented as the correlation coefficient (*r*) with the corresponding significance.

## Data Availability

The data are available upon reasonable request to the Corresponding Author.

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
