# Peer review of "Self-Reported Musculoskeletal Disorder Symptoms among Bus Drivers in the Taipei Metropolitan Area"

_ijerph, 2022, doi:10.3390/ijerph191710596_

Round 1

Reviewer 1 Report

Based on the NMQ questionnaire and logistic regression analysis, this study investigated the musculoskeletal disorders of bus drivers working in Taipei metropolitan area, analyzed the prevalence of various parts of the body and related risk factors, and put forward prevention suggestions, providing reference for musculoskeletal disorders research in different regions and regions. It has research significance and rationality. The experimental and analytical methods of this study are highly applicable and feasible, but the author can increase the explanation of the revision of the NMQ questionnaire and the reasons for the revision, so that other researchers can have enough details to replicate the proposed experimental procedures and analysis. When comparing the musculoskeletal discomforts of various parts of the body of workers in different industries, it can increase the reliability of the data of various industries, the number of samples in the research results and the elaboration of the comparison conclusions. The author can increase the relevant analysis and discussion of the questionnaire results and enhance the readability of the form.

Reviewer 2 Report

Dear Authors,

your study entitled "Self-Reported Musculoskeletal Disorder Symptoms among Bus Drivers in the Taipei Metropolitan Area" can be interesting, but it needs major revisions before it can be considered for possible publication in IJERPH.

From a scientific point of view, the design of your study is very poor, as it is a cross sectional analysis of a convenience sample (any information on informed consent? ethical approval?). All the information you collected were self-reported by the subjects, with the exception of the 5 (why only 5?) interviews, which I did not fully understand how they were used for the analysis.

These limitations have to be fully discussed ibn your paper, and I hope that some of these points can be improved with a further version of your article.

Moreover, there are also methodological errors in writing the paper, as you misleadingly reported Tables of results describing your population in the methodological section, and you also reported results that you did not find, but based on scientific literature, in a Table placed in the Results section instead of the Discussion section.

The main strengths of your study that I can see is that:

1. You addressed a population which is still not fully addreesed in all the parts of the world, i.e. bus driving, and ii can be interested to further evaluate it because driving, vehicles and roads conditions can be very different, significantly influencing WRMSd: please take properly into account in the discussion also the role of vibrations, as reported e.g. in this paper, that can be cited to add more recent references to your article, that has quite old references: DOI: 10.3233/WOR-205007

2. You used a widely applied questionnaire, as the NMQ questionnaire, so you have a lot of data to compare with the data you obtained, considering both the prevalence of the symptoms, as well as the associations of these symptoms with different potential risk factors among bus drivers. Please further elaborate on this in your discussion and have a much more detailed look at the studies currently published in scientific literature applying the NMQ in professional drivers and at the results the other Authors obtained, also considering the setting and the place where the study was conducted, to see whethere the results are in accordance or not with yours. Here is a list of seven articles on professional drivers and musculoskeletal symptoms: please have a look at them and consider to discuss some of them (if not already included in your study) and compare their results with your results:

doi: 10.1093/occmed/kqi125. 

- doi: 10.2105/ajph.94.4.575.

doi: 10.1007/s00420-021-01683-1

doi: 10.1080/10803548.2018.1433107

doi: 10.1080/10803548.2021.1952773

doi: 10.1080/10803548.2020.1834233

doi: 10.1093/occmed/52.1.4

One of the main problem that I found in your article is that, as you probably know, the NMQ has two different questions to evaluate WRMSD. The first is "Have you at any time during the last 12 months had trouble (such as ache, pain, discomfort numbness) in", while the second is "During the last 12 months have you been prevented from carrying out normal activities (e.g. job, housework, hobbies) because of this trouble in". The problem with using the first question, as you did and also the majority of other researchers do, is that you find very very high prevalences of symptoms, but it is diffuclt to understand whether these symptoms are really work related, because if you ask the same question to the general public, to young subvjects, to people with no particular exposure to occupational factors determining WRMSDs you can obtain very high prevalences too. For this reason, it would be important to report also the prevalences of the ansers to the second question, that, from one side, can underestimate the real prevalence of WRMDs, but. on the other hand, is much more precise in telling whethere the symptoms can interfere with the job of the subjects. Is there any chance for you to get this data, as the number of people investigated is kind of low, and report the prevalence obtained with the answers to the second question?

Finally, let me mention also a final point, that I did not find fully discussed in your paper: bus drivers are exposed to a high level of work stress, due to the risk of accidents, the traffic, the relations with the bus users. The perceived stress level can influence the prevalence of muscoloskeletal symptoms, as well as the risk of injuries and the personality of bus drivers. You can find some interesting data in this paper that can be cited in your article: doi: 10.3233/WOR-172581 . Please elaborate also on these consepts in your revised discussion.

With best regards,

the Reviewer

Round 2

Reviewer 2 Report

Congratulations to the Authors as they properly addressed all of the Reviewers' comments and suggestions. My felling is that the article can be considered ready for publication now.

Best regards,

the Reviewer